# Immune-Cell-Derived Exosomes as a Potential Novel Tool to Investigate Immune Responsiveness in SCLC Patients: A Proof-of-Concept Study

**DOI:** 10.3390/cancers16183151

**Published:** 2024-09-14

**Authors:** Luisa Amato, Caterina De Rosa, Viviana De Rosa, Hamid Heydari Sheikhhossein, Annalisa Ariano, Paola Franco, Valeria Nele, Sara Capaldo, Gaetano Di Guida, Filippo Sepe, Alessandra Di Liello, Giuseppe De Rosa, Concetta Tuccillo, Antonio Gambardella, Fortunato Ciardiello, Floriana Morgillo, Virginia Tirino, Carminia Maria Della Corte, Francesca Iommelli, Giovanni Vicidomini

**Affiliations:** 1Department of Precision Medicine, University of Campania Luigi Vanvitelli, 80131 Naples, Italyannalisaariano9826@gmail.com (A.A.); saracapaldo99@gmail.com (S.C.); gaetano.diguida@studenti.unicampania.it (G.D.G.); filippo.sepe@studenti.unicampania.it (F.S.); alessandradiliello@gmail.com (A.D.L.); concetta.tuccillo@unicampania.it (C.T.); antonio.gambardella@unicampania.it (A.G.); fortunato.ciardiello@unicampania.it (F.C.); floriana.morgillo@unicampania.it (F.M.); 2Institute of Biostructures and Bioimaging, National Research Council, 80145 Naples, Italy; viviana.derosa@ibb.cnr.it (V.D.R.); francesca.iommelli@ibb.cnr.it (F.I.); 3Department of Medical, Oral and Biotechnological Sciences, University “G. d’Annunzio” of Chieti-Pescara, 66100 Chieti, Italy; hamid.heydari@unich.it; 4Villa Serena Foundation for Research, 65013 Città Sant’Angelo, Italy; 5Institute of Genetics and Biophysics Adriano Buzzati Traverso, National Research Council, 80131 Naples, Italy; paola.franco@igb.cnr.it; 6Department of Pharmacy, University of Naples Federico II, 80131 Naples, Italy; valeria.nele@unina.it (V.N.); gderosa@unina.it (G.D.R.); 7Department of Experimental Medicine, University of Campania Luigi Vanvitelli, 81100 Caserta, Italy; 8Department of Translational Medical Sciences, University of Campania Luigi Vanvitelli, 80131 Naples, Italy

**Keywords:** biomarkers, exosomes, PBMCs

## Abstract

**Simple Summary:**

In the era of precision medicine and immunotherapy, the isolation and characterization of exosomes from the peripheral blood mononuclear cells (PBMC-EXs) of SCLC patients may represent a new tool to define responder (BR) from non-responder (NR) patients undergoing chemoimmunotherapy treatment. In this proof-of-concept study, we isolated PBMC-EXs from the peripheral blood of SCLC patients and investigated the potential role of such extracellular vesicles (EVs) in monitoring tumor response to drug stimuli. Interestingly, we found increased exosome levels of c-Myc and Snail along with reduced levels of the immune markers MAVS and STING in NR patients. Also, we showed that PBMC-EXs from BR patients induced an increase in apoptosis and a reduction in the cell viability of SCLC cells compared to PBMC-EXs from NR SCLC patients. Thus, we suggest that PBMC-EXs may represent an innovative strategy to be further explored for the therapy and selection of immune-responsive SCLC patients.

**Abstract:**

Small cell lung cancer (SCLC) is a highly invasive and rapidly proliferating lung tumor subtype. Most patients respond well to a combination of platinum-based chemotherapy and PD-1/PDL-1 inhibitors. Unfortunately, not all patients benefit from this treatment regimen, and few alternative therapies are available. In this scenario, the identification of new biomarkers and differential therapeutic strategies to improve tumor response becomes urgent. Here, we investigated the role of exosomes (EXs) released from the peripheral blood mononuclear cells (PBMCs) of SCLC patients in mediating the functional crosstalk between the immune system and tumors in response to treatments. In this study, we showed that PBMC-EXs from SCLC patients with different responses to chemoimmunotherapy showed different levels of immune (STING and MAVS) and EMT (Snail and c-Myc) markers. We demonstrated that PBMC-EXs derived from best responder (BR) patients were able to induce a significant increase in apoptosis in SCLC cell lines in vitro compared to PBMC-EXs derived from non-responder (NR) SCLC patients. PBMC-EXs were able to affect cell viability and modulate apoptotic markers, DNA damage and the replication stress pathway, as well as the occurrence of EMT. Our work provides proof of concept that PBMC-EXs can be used as a tool to study the crosstalk between cancer cells and immune cells and that PBMC-EXs exhibit an in vitro ability to promote cancer cell death and reduce tumor aggressiveness.

## 1. Introduction

Small cell lung cancer (SCLC) is a highly invasive and rapidly proliferating pathologic subtype that accounts for 13–15% of all lung cancer cases [1]. For decades, platinum-based chemotherapy regimens have constituted the single option of therapy for patients with SCLC, with a five-year overall survival (OS) rate of only 10% [2,3,4]. Nowadays, although the introduction of inhibitors of programmed cell death protein-1/programmed death-ligand 1 (PD-1/PD-L1) in combination with chemotherapy as a first-line treatment has led to a significant improvement in OS and progression free survival (PFS) [5], not all patients with SCLC benefit from this treatment schedule, and few alternative therapies are available for patients. The mechanisms behind the lack of response to immunotherapy remain unclear. Both factors intrinsic (i.e., lack of tumor antigens, impaired antigen presentation, genetic T-cell exclusion) and extrinsic (i.e., lack of T cells, inhibitory immune checkpoints, immunosuppressive cells) to the tumor cell play a role in immune resistance [6]. In this regard, the identification of baseline markers in the plasma samples of patients who will benefit most from a chemoimmunotherapy regimen remains an important challenge. The biomarker-driven categorization of best responders (BRs) and non-responders (NRs) would increase the success rate of therapies and further advance personalized treatments [7]. Therefore, there is an urgent clinical need to improve therapeutic options and identify new biomarkers for predicting the immunotherapy response of candidate patients to different therapeutic strategies. Although several advances in the understanding of antitumor immune response have been made, and it is recognized as playing a crucial role in the SCLC tumor microenvironment (TME) in mediating this process, the interaction between TME and cancer cells is not completely clear and difficult to investigate in a clinical setting. Despite restrictions in representing the TME, circulating blood-based cellular biomarkers offer significant advantages over tumor tissue in terms of sample accessibility, the possibility of quantitative measurement and longitudinal monitoring, as well as access to dedicated systems for analysis [8]. Recently, the blood immune cell signature has emerged as a novel tool to monitor the treatment with chemoimmunotherapy, thus contributing to a better prediction of SCLC patient outcomes [9]. In addition, the use of noninvasive and dynamic approaches such as body fluid analysis may also be very easily moved to the clinic. Recently, our group and others have demonstrated the potential role of circulating peripheral immune cells (PBMCs) to investigate the activation of antitumor immune response [10,11,12], but unfortunately, it is not well understood which actors are involved in this process. Among the circulating blood-based biomarkers, exosomes isolated from immune cells are being considered as potential immunotherapeutic reagents because of their ability to modulate the immune response. In fact, it has been shown that both tumor and immune-cell-derived exosomes can deliver tumor antigens and boost immunity [13,14], with no data available for SCLC patients. In this respect, we focused our attention on the role of PBMC-derived exosomes (PBMC-EXs) in mediating the crosstalk between the immune system and tumors in SCLC. Exosomes are highly heterogeneous bilayer vesicles with a diameter of approximately 30–200 nm [15,16] that can be secreted by many types of cells, including cancer cells and immune cells [17]. They may be isolated from cell cultures and biological fluids and may carry in the extracellular space mRNA, DNA, lipids and proteins. In particular, the composition and properties of exosomes may be affected by different cell types and upon drug stimulation. However, some common intraluminal markers such as chaperone proteins and membrane antigens such as tetraspanins (CD9, CD81, CD63, CD82) have been identified in such vesicles. Although they were initially considered to be a cellular waste, it is now well recognized that exosomes represent carriers for cell-to-cell communication and contribute to a wide range of biological processes in both physiological and pathological conditions, including cancer [18,19]. In particular, exosome internalization in recipient cells is able to modulate several intracellular signal cascades involved in proliferation, cell death and response to treatment [17,20].

In this work, we hypothesized that exosomes isolated from SCLC-patient-derived PBMCs (PBMC-EXs) may predict different tumor responses to chemoimmunotherapy. Thus, we designed a proof-of-concept study to shed light on the role of PBMC-EXs in this clinical context. We aimed to isolate PBMC-EXs from the peripheral blood of R and NR patients in chemoimmunotherapy and investigate them alongside tumor resistance or sensitivity to treatment. Also, we planned to assess their potential ability to exert in vitro cytotoxic effects on SCLC cells.

## 2. Materials and Methods

### 2.1. Study Population, Patient Characteristics and Clinical Responses

In this study, 10 patients with a diagnosis of limited stage (LD) or extensive stage (ES) SCLC who had undergone at least two cycles of PD-L1 inhibitors between 2022 and 2023 were enrolled. We enrolled patients with a diagnosis of SCLC receiving one of the following treatments: chemotherapy (cisplatin) and/or an anti-PD-L1 antibody (atezolizumab, durvalumab). The clinical and demographic variables for lung cancer cases are detailed in Table 1. The patients were grouped as follows: best responders (BRs) were considered as patients with a controlled disease of SCLC lasting more than 6 months that were clinically relevant compared to registration clinical trials [21], while non-responders (NRs) were defined as patients showing progression of disease (PD) as the best clinical response.

### 2.2. Cell Lines

Human small cell lung cancer cell lines NCI-H446 (ATCC Cat#HTB-171; RRID: CVCL_1562) and NCI-H661 (ATCC Cat#HTB-183; RRID: CVCL_1577) were purchased from the American Type Culture Collection (ATCC). All cells were cultured in an RPMI-1640 medium (Sigma-Aldrich, R8758, St. Louis, MI, USA) supplemented with 10% FBS (Sigma-Aldrich) and 1× penicillin–streptomycin (Sigma-Aldrich, P0781) and incubated in a humidity-controlled environment (37 °C, 5% CO_2_). All cell lines were routinely tested to exclude mycoplasma contamination using a mycoplasma detection kit (InvivoGen, San Diego, CA, USA).

### 2.3. Isolation of Peripheral Blood Mononuclear Cells (PBMCs)

Human samples were collected after obtaining a written informed consensus from patients in accordance with the Declaration of Helsinki. The protocol for the use of these samples for research purposes was approved by the Ethics Committee of the University of Campania “Luigi Vanvitelli”, Naples (n. 280 on 16 May 2020). For the isolation of exosomes from patient-derived PBMCs (PBMC-EXs), lung cancer patients’ blood was collected in BD vacutainer spray-coated K2EDTA tubes (BD, Franklin Lakes, NJ, USA) as described in our previous studies [10]. Briefly, PBMCs were isolated by using Lymphosep (Aurogene, Rome, Italy) gradient centrifugation. Furthermore, to eliminate the contamination of red blood cells (RBCs) in the PBMC samples, we suspended the PBMC pellets in an RBC lysis solution (Invitrogen™, Waltham, MA, USA). After three washes with PBS, PBMCs were cultured in an RPMI-160 medium (Sigma-Aldrich, R8758) supplemented with 10% of exosome-free FBS and 1× penicillin–streptomycin (Sigma-Aldrich, P0781). After 72 h, the medium was collected to start the exosome isolation. Exosome-free FBS was obtained by ultracentrifugation at 100,000× *g* at 4 °C for 16–18 h [22]. The pellet was discarded, and the supernatant was filtered through a 0.22-μm PES filter (Millipore, Burlington, MA, USA) [23].

### 2.4. Exosome Isolation and Characterization from PBMCs

The medium containing PBMCs was centrifuged at 300× *g* for 12 min to pellet the cells [24]. Subsequently, the supernatant was centrifuged at 2000× *g* for 10 min to remove dead cells. The exosome-containing medium was transferred to ultracentrifuge tubes (Beckman Ultra-clear tubes, Beckman Coulter, Brea, CA, USA). The ultracentrifuge (SW 41 rotor, swinging bucket) was set at 10,000× *g* for 30 min at 4 °C. After this time, the pellet was discarded, and to increase the purity of samples [23], the supernatant containing exosomes was filtered through a 0.22 μm filter (Millipore) and transferred to new tubes and centrifuged at 4 °C and 100,000× *g* for 70 min. The pellet was washed in cold PBS, and an additional centrifuge at 100,000× *g* for 70 min was performed. The supernatant was removed, and the pellet (exosomes) was resuspended in 100–200 µL of cold sterile PBS. The exosomes from PBMCs (PBMC-EXs) were then stored at −80 °C for further use. The schematic process of exosome isolation is shown in Figure 1. After isolation, the exosomes were characterized by nanoparticle tracking analysis (NTA) to detect the size and particle number in the selected samples. Images using electron microscopy (AlfaTest Instrument, Timisoara, Romania) were also captured to visualize the isolated exosomes. Exosome protein content was determined by a Bradford assay (Bio-Rad, Hercules, CA, USA) as previously described [25,26,27,28,29,30,31,32,33,34,35].

### 2.5. Exosome Characterization by Scanning Electron Microscopy (SEM) and Nanoparticle Tracking Analysis (NTA)

Pellets containing PBMC-EXs were resuspended in 0.1–0.4 mL of PBS, and a few microliters of suspension were deposited on a cover glass and dried overnight. The samples were positioned on a stub, directly transferred to an SEM (FEG SEM model Pharos from Thermo Fisher Scientific, Waltham, MA, USA) and subsequently sputter coated with an automatic sputter (Luxor Pt coater model, APTCO TECHNOLOGIES, Nazareth, Belgium) using gold in air and setting a thickness of approximately 5 nm. The samples were then analyzed in a high vacuum using an SED (secondary electron) detector at an accelerating voltage of 15 kV. At least 3 EM images were acquired for each sample.

Nanoparticle tracking analysis (NTA) was performed to measure the size and particle concentration of PBMC-EXs using a Malvern Panalytical NanoSight Pro (Malvern Instruments, Amesbury, UK) equipped with a 488 nm laser. Samples were vortexed and diluted at 1000× or 2000× in 0.2 µm filtered 1X PBS prior to the analysis. A high-sensitivity sCMOS (USB-3) camera was used to acquire five videos with software-optimized settings and a sample flow rate between 5 and 15 μL/min; the videos were then analyzed with NS Xplorer software v1.1.0.6 to obtain the size and particle concentration of the samples.

### 2.6. Co-Culture Protocol

SCLC cells H446 or H661 were seeded at 500,000 cells/well in a 6-well plate and cultured in a culture medium containing PBMC-EXs (25 μg) or PBS as the control. SCLC cells with and without PBMC-EXs were incubated for 24 h or for 72 h at 37 °C in a humidified incubator of 5% CO_2_.

### 2.7. Western Blot Analysis

SCLC cells were lysed by homogenization in an RIPA lysis buffer [0.1% sodium dodecyl sulfate (SDS), 0.5% deoxycholate, 1% Nonidet, 100 mmol/L NaCl, 10 mmol/L Tris–HCl (pH 7.4), 0.5 mmol/L dithiothreitol and 0.5% phenylmethyl sulfonyl fluoride], protease inhibitor cocktail (Hoffmann-La Roche, Basel, Switzerland) and phosphatase inhibitor tablets (PhosSTOP; Roche Diagnostics, Basel, Switzerland) and clarification by centrifugation at 2348 rcf for 20 min at 4 °C. Whole cell lysates or PBMC-EX samples containing comparable amounts of proteins were resuspended in LDS reducing sample buffer, mixed and boiled at 100 °C for 10′. Samples were resolved by SDS-PAGE gels and electrotransferred onto 0.2 µm nitrocellulose membranes (Trans-Blot Turbo; BioRad). After blocking membranes for 90 min at room temperature, they were incubated overnight at 4 °C with primary antibodies and then with a secondary antibody for 1 h at room temperature. Horseradish peroxidase-linked anti-rabbit (BioRad) and anti-mouse (BioRad) antibodies were used as secondary antibodies. Proteins were detected with a Clarity Western ECL Substrate using the ChemiDoc system (BioRad). Images were analyzed using BioRad software Image Lab 3.0.1. The primary antibodies for Western blot analysis of STING (D2P2F) (13647, 1:1000), MAVS (D5A9E) (24930, 1:1000), e-cadherin (24E10) (3195, 1:1000), Snail (L70G2) (3895, 1:1000), c-myc (D84C12) (5605, 1:1000), BCL-XL (54H6) (2764, 1:1000), Bcl-2, BID (2002, 1:1000), Caspase-8 (1C12) (9746, 1:1000), Bcl-2 (2872, 1:1000), lamin A/C (2032, 1:1000), Phospho-Chk2 (Thr68) (C13C1) (2197, 1:1000), Chk2 (D9C6) (6334, 1:1000), Histone H2A.X (D17A3) (7631, 1:1000), phospho-p44/42 MAPK (Erk1/2) (Thr202/Tyr204) (9101, 1:1000), p44/42 MAPK (Erk1/2) (9102, 1:1000), TGFBR-I (3712, 1:1000), GAPDH (D16H11) (5174, 1:1000) and α-Tubulin (DM1A) (3873, 1:1000) were purchased from Cell Signaling (Danvers, MA, USA). Monoclonal anti-HSP70 (sc-24, 1:1000), anti-CD81 (sc-166029) and anti-CD63 (sc-15363) antibodies were obtained from Santa Cruz Biotechnology, Dallas, Germany. Monoclonal anti-Calnexin (SPA-860) was purchased from Stressgen Biotechnologies, Victoria, BC, Canada.

### 2.8. MTT Assay

The PBMC-EX-induced toxicity of SCLC cells was assessed using an MTT assay. Briefly, SCLC cells were seeded in 96-well flat-bottomed plates at a density of 3000 cells/well and cultured in a culture medium containing the PBMC-EXs (co-culture) or PBS (control). After 24 h or 72 h of co-culture with PBMC-EXs, the methylthiazolyldiphenyl-tetrazolium bromide (MTT, Sigma Chemical Co., St. Louis, MO, USA) was added (10 μL/well) to assess its metabolization to formazan salt. After 4 h, cells were lysed by the addition of DMSO. The number of viable cells was determined spectrophotometrically by measuring absorbance at 490 nm and expressed as the percentage of viable cells, considering the untreated control cells as 100%. At least three independent experiments were performed in triplicate.

### 2.9. Apoptotic Assay

At the end of the co-culture incubation for 24 or 72 h, cells were collected, washed with PBS and resuspended in a 1X annexin-binding buffer. Cells were then stained with Annexin V-Alexa 488 conjugate and PI according to the manufacturer’s protocol (“Invitrogen™ Alexa Fluor™ 488 annexin V/Dead Cell Apoptosis Kit”, cat.no V13241) and analyzed by flow cytometry using a BD Fortessa (BD Biosciences, Franklin Lakes, NJ, USA). Analysis was conducted using BD FACSDiva™ Software, version 8.0 (BD Biosciences).

### 2.10. Statistical and Image Analysis

Results were expressed as the mean ± SEM or SD. Three or more groups with one independent variable were analyzed using a one-way ANOVA test. Analyses were performed using Prism 8 (GraphPad Software, San Diego, CA, USA) software. All tests were two-tailed, and a *p*-value < 0.05 was considered to indicate statistical significance. All the experiments were repeated a minimum of three times independently to ensure reproducibility.

The Western blot signals were quantified by morpho-densitometric analysis using ImageJ software version Java8 (NIH, Bethesda, MD, USA). Briefly, the product of the area and optical density of each band was determined and normalized to the same parameter derived from the equal loading used. The data were expressed as the relative protein levels of each sample compared with those of the corresponding equal loading.

## 3. Results

### 3.1. PBMC-Derived Exosome (PBMC-EX) Characterization

Exosomes were isolated from PBMCs derived from SCLC patients via a differential ultracentrifugation technique. Imaging via scanning electron microscopy (SEM) was performed to visualize morphology and assess exosome size (Figure 2A). In addition to these data, a size distribution analysis was performed by nanoparticle tracking analysis (NTA), which revealed a homogeneous exosome population for both BR and NR PBMC-EXs. In particular, as reported in the size plots (Figure 2B), most isolated extracellular vesicles (EVs) showed a size range < 200 nm in diameter, and accordingly with the minimal information for studies of extracellular vesicles (MISEV2023) guidelines [36], exosome diameters were included in such a range. However, due to the presence of a low fraction of EVs from both NR and R patients, with a diameter of or higher than 200 nm, we supposed that our samples also contained a small percentage of microvesicles, possibly due to the heterogeneity of PBMC populations. The data presented in Appendix A were obtained from the NTA analysis and showed the mean size of all isolated EVs from both NR and R patients.

In addition, Western blot analyses were performed to further characterize the exosomes. In particular, CD63 and CD81, two well-known specific exosomal markers of the Tetraspanins family [37], were expressed in isolated PBMC-EXs (Figure 2C), whereas the Calnexin (a known negative marker for exosomes) was selectively expressed only by SCLC cells and by donor PBMCs from SCLC patients, and it was absent in PBMC-EXs, thus confirming the identity of PBMC-EXs (Appendix A).

### 3.2. Innate Immune DNA/RNA Sensors and EMT TFs Are Differentially Carried in PBMC-EXs

To better characterize the biological properties of purified exosomes and relevant proteins known for affecting cancer immune responses, a Western blot analysis was performed. First, HSP70 was highly and equally expressed in BR and NR PBMC-EXs and, therefore, was used as an internal equal loading (Figure 2C).

We recently demonstrated that cGAS/STING activation in PBMCs represents a useful tool to predict the response to immunotherapy in LC patients [4]. As we used PBMCs as a source of exosomes, we investigated the expression of innate immune DNA/RNA sensors in isolated PBMC-EXs. Interestingly, we found that the PBMC-EXs of BR patients expressed higher levels of innate immune sensors, namely, the stimulator of interferon genes (STING) and mitochondrial-antiviral signaling (MAVS) proteins, compared to the PBMC-EXs of NR patients (1.26- and 1.60-fold increase, respectively) (Figure 2C). These findings provide the first evidence that PBMC-EXs may act as carriers of innate immune DNA/RNA sensors but also that this peculiarity reproduces the PBMC source phenotype, thus reflecting the immune activation in response to therapy in SCLC patients. Furthermore, based on the recent evidence showing that STING pathways in cancer cells may be also involved in cell death induction [38,39,40], we believe that an effective delivery of STING and MAVS through PBMC-EXs may enhance the suppression of tumor growth and activate an antitumor immunity response.

Given the critical role of c-Myc and EMT signaling in SCLC biology, we also sought to characterize PBMC-EXs for their ability to carry such oncogene and transcription factors (TFs) highly involved in EMT activation. As reported in Figure 2C, we detected a lower expression of c-MYC and Snail in the PBMC-EXs isolated from BR patients compared to the PBMC-EXs from NR patients (0.06- and 0.02-fold decrease, respectively). We speculate that it is possible that whether released in an in situ TME, PBMC-EXs from BR patients may modulate tumor cells by promoting immunological cell death. Moreover, they may act as shuttles for immune cells to carry transcription factors (TFs) strictly involved in the reprogramming of cancer cell phenotypes.

### 3.3. PBMC-EXs Induce Cell Death in Co-Culture with SCLC Cells

Recently, the number of exosomes released by tumors has been proposed as a novel tool for early-stage cancer detection [41]. So, we wondered whether clinical features, like SCLC stage and patient response to therapy may impact the type and amount of proteins contained in PBMC-EXs. We measured the yield of proteins (µg) obtained from PBMC-EXs isolated from patients with the diagnosis of limited stage (LS)-SCLC (n = 2), extensive stage (ES)-SCLC BR (n = 2) and (ES)-SCLC NR (n = 2). In particular, we isolated approximately 20 × 10^6^ ± 10 PBMCs from 12 mL of blood samples from patients. As shown in Table 2 and in our experimental conditions, the amount of proteins in the samples from (LS)-SCLC patients was under the detection limit, whereas we successfully detected the protein concentration in PBMC-EXs from (ES)-SCLC patients. Interestingly, the protein amounts from PBMC-EXs derived from NR patients were higher than the amount obtained from PBMC-EXs isolated from BR patients. These findings need to be validated in terms of the number of PBMC-EXs and in a larger cohort of patients, as it suggests that tumor stage and/or response to therapy may differentially affect PBMC-EX release in SCLC patients.

Given the widely documented role of DNA/RNA sensors like cGAS-STING and MAVS in facilitating diverse cell death pathways in antitumor immune response [42,43], we aimed to determine whether PBMC-EXs from BR versus NR patients, carrying different levels of STING and MAVS, could differentially affect cell viability in SCLC cell lines. To this aim, we selected two different SCLC cell lines, the first being H661 cells, which are characterized by growth in an adherent condition, exhibiting an epithelial phenotype and no macroscopic DNA structural abnormalities. The second was H446 cells, which are characterized by growth in a mixed-culture condition (adherent and suspension) and a high mesenchymal phenotype, including amplification of the MYC DNA sequence.

We performed a co-culture of PBMC-EXs isolated from both BR and NR SCLC patients with H661 and H446 SCLC cell lines and evaluated SCLC cell viability by performing an MTT assay. We assessed early and late apoptosis by Annexin V/PI staining as an early (after 24 h of co-culture) and late response (after 72 h of co-culture). We found that PBMC-EXs obtained from BR and NR SCLC patients were able to induce a significant reduction in cell viability in both H661 and H446 cell lines after 24 h of co-culture. Statistical significance was considerably higher when SCLC cell lines were co-cultured with PBMC-EXs from BR patients (Figure 3A). Interestingly, we also found that PBMC-EXs from BR SCLC patients were able to significantly reduce cell viability in both H661 (27.7% cell death, *p* < 0.0001) and H446 (31.8% cell death, *p* < 0.0001) SCLC cell lines after 72 h of co-culture. Conversely, at the same time point, PBMC-EXs from NR SCLC patients caused a weak reduction in cell viability in H661 cells (7% cell death, *p* < 0.01) and did not have any significant effect on the cell viability in H446 cells (Figure 3B). These results suggest that PBMC-EXs from both BR and NR patients are able to mediate a significant early response of SCLC cells in terms of a reduction in cell viability, while PBMC-EXs from BRs are able to promote a long-term reduction in cell viability compared to NR PBMC-EXs.

In parallel, in order to perform a more accurate assessment of cell death, we performed Annexin V/PI apoptosis staining via a flow cytometry assay. After 24 h of incubation, we found in H661 cells a higher cell percentage in the phase of late apoptosis than in the phase of early cell death. Similar results were obtained for H661 cells co-cultured with both BR PBMC-EXs (67.6%) and NR PBMC-EXs (44.5%). Conversely, in H446 cells, we found a higher percentage of early apoptosis in the co-culture with both BR PBMC-EXs (48.8%) and NR PBMC-EXs (46.1%) (Appendix A) in comparison with the percentage of cells in the late phase of cell death. Different results were obtained after 72 h of co-culture. In particular, we found that PBMC-EXs from BR SCLC patients caused a similar increase in late apoptosis in both cell lines (H661: 31.2% and H446: 23.4%), while early apoptosis was only increased in H661 cells (27.8%) compared to H446 cells (5.9%) (Figure 4A,B). Moreover, co-culture did not affect either early or late apoptosis in both the cell lines of NR PBMC-EXs compared to the PBS control. These results confirmed the MTT assay findings, showing that PBMC-EXs from BR SCLC patients are able to sustain SCLC cell death for a longer time compared to PBMC-EXs from NR SCLC patients.

Taken together, these findings suggest that PBMC-EXs derived from SCLC patients are able to induce in vitro cell death. In addition, the in vitro cytotoxicity is dependent on the exposure time of SCLC cells to PBMC-EXs and based on the different biological properties and phenotypes of PBMC-EXs depending on the SCLC patient’s response to therapy; the two selected cell lines showed a different response in terms of cell viability and apoptosis.

### 3.4. Effect of PBMC-EXs from BR and NR Patients on the Modulation of Cell Death Mediators, DNA Damage Pathways and EMT Signaling

In order to further understand the effects of BR and NR PBMC-EXs in co-cultured SCLC cells, whole cell lysates from exosome-stimulated and unstimulated cells were analyzed. To this end, the levels of signaling mediators and TFs involved in cellular stress (DNA damage), apoptosis and EMT pathways were analyzed as a surrogate of cellular proliferation.

We selected PBMC-EXs isolated from both BR and NR SCLC patients and performed a co-culture with H661 and H446 SCLC cells. After 72 h of incubation, protein extraction and Western blot analysis were performed. In particular, in H661 cells, PBMC-EXs from BR patients were able to induce a strong upregulation of the initiator caspase 8 that is involved in the activation of the extrinsic apoptotic pathway, thus promoting apoptosis by proteolytic processing and activation of executioner caspases (Figure 5A). In agreement with these results, we also detected that BR PBMC-EXs caused a concomitant cleavage of lamin A/C, a nuclear protein that when present in its cleaved form represents an end-point marker of the apoptosis cascade [44]. In the same cell line, NR PBMC-EXs were able to induce lamin A/C cleavage but not upregulation of caspase 8. The co-culture with both BR and NR PBMC-EXs reduced tBID expression in the H661 cell line. Further studies will be necessary to better elucidate the mechanisms of BR PBMC-EX-mediated cell death in H661 cell lines. In H446 SCLC cells co-cultured with BR PBMC-EXs, upon apoptotic marker screening, we found an increase in tBID, an effector of mitochondrial permeabilization during apoptosis associated with cytochrome c release. Opposite results were obtained in H446 SCLC cells co-cultured with NR PBMC-EXs (Figure 5B). These findings indicated that BR PBMC-EXs were able to induce both an extrinsic and intrinsic apoptotic pathway activation depending on cell line phenotype that requires further studies to deeply understand the molecular mechanisms.

Moreover, we characterized the selected SCLC cell lines after co-culture with PBMC-EXs from BR and NR SCLC patients after 72 h of co-culture in terms of DNA damage and replication stress markers, two hallmarks often amplified in SCLC tumors. As shown in Figure 5A, H661 cells co-cultured with BR PBMC-EXs showed an increase in Bcl-2/Bcl-xL along with a reduction in p-Chk2/Chk2, thus suggesting the intrinsic cell phenotype may affect these responses. Conversely, Bcl-xL expression was selectively increased in H661 cells co-cultured with NR PBMC-EXs.

Furthermore, as shown in Figure 5B, we also observed that BR PBMC-EXs were able to increase the levels of DNA damage and replication stress markers in H446 SCLC cells, with a significant increase in Bcl-2, Bcl-xL and H2A.X. In the same cell line, NR PBMC-EXs induced no significant change in Bcl-2 and H2A.X and only a slight increase in Bcl-xL. Conversely, the levels of phosphorylated and total Chk2, in the co-culture with PBMC-EXs from both BR and NR patients, were reduced in H446 cells. Interestingly, we also found, in H446 cells, that increases in cell death and intracellular stress upon co-culture with PBMC-EXs from BR SCLC patients were also associated with a transition toward a less aggressive cell phenotype, as demonstrated by a reduction in MAPK and TGFBR-I. In addition, similar to apoptotic markers, PBMC-EXs were able to determine a different effect on EMT marker expression in two SCLC cell lines. A reduction in Snail, phospho- and total MAPK, along with TGFBR-I, was observed in H661 cells co-cultured with PBMC-EXs from both BR and NR patients. In the H446 cell line, we observed a reduction in the mesenchymal markers Snail, MAPK and TGFBR-I, alongside an increase in E-cadherin levels, in the co-culture with BR PBMC-EXs. Conversely, in H446 cells co-cultured with NR PBMC-EXs, we detected an increase in TGFBR-I and E-cadherin. Taken together, these results indicate that the balance between proliferation and apoptosis is differentially perturbed by PBMC-EXs depending on the patient source and the heterogeneity related to the response to therapies. A semiquantitative analysis of the results shown in Figure 5A,B is shown in Appendix A.

In summary, our data indicated that, based on their different protein expressions, BR and NR PBMC-EXs differently affect the levels of apoptotic markers, DNA damage/replication stress pathways and EMT signaling in SCLC cell lines. Furthermore, the exosome-dependent intracellular response is also affected by the different cellular phenotypes and biological properties of the SCLC cells selected for this study.

## 4. Discussion

SCLC represents one of the most difficult cancers to treat due to the lack of well-defined biomarkers and multiple drug targets useful to design a personalized efficient therapy in oncology. Also, diagnosis and biomarker studies are affected by the little amount of available tumor tissue, with no targeted agents available. Currently, given the drug combination of chemo- and immunotherapy being the main treatment option, it is known that the immune response is heterogeneous, with only one subgroup of patients (inflamed subtype) showing a high benefit from immunotherapy treatment. In this respect, research is needed to identify the best responders (BRs) to chemoimmunotherapy in clinical practice.

Recently, several studies have demonstrated the impact of exosomes on the innate immune system and response to immunotherapy in lung cancer patients [45,46]. However, the effects of exosomes released from the PBMCs of SCLC patients are currently unknown. For the present study, we collected serially PBMCs from SCLC patients, and we identified the BR and NR SCLC patients in chemoimmunotherapy based on the clinical response to therapy. From these patients, we isolated exosomes from PBMCs (PBMC-EXs), for which we analyzed the expression of selected proteins and assessed the cytotoxic ability when co-cultured with SCLC cells in vitro. We found that, in comparison with NRs, the PBMC-EXs from BR patients showed lower levels of the EMT transcriptional factors (TFs) Snail and c-Myc and higher levels of the antitumoral innate immune DNA/RNA sensors STING and MAVS. These findings confirm and reinforce our previous study demonstrating that PBMCs from BR lung cancer patients showed the highest levels of STING pathway expression along with an increased tumor infiltration ability of immune cells [10]. In addition, since EMT represents a hallmark of SCLC [47,48,49], the increased EMT TF expression in PBMC-EXs from NR patients may be involved in the crosstalk between PBMC-EXs and cancer cells and modulate tumor response. These results suggest that PBMC-EXs may be different between resistant and sensitive patients and may be tested in further extended studies as clinical biomarkers of therapy response. In addition, we tested the effect of PBMC-EXs on SCLC cells and found that the exosomes isolated from BR patients were able to exert in vitro cytotoxicity and increase cell death as a long-term response. In particular, after 24 h of incubation, PBMC-EXs were observed to induce cell apoptosis and reduce cell viability, independent of the origin of the cohort of patients. Following 72 h of incubation, PBMC-EXs derived from BR patients induced cell apoptosis in comparison to NR patients. Furthermore, a panel of markers associated with apoptosis, DNA damage and replication stress with implications for EMT was screened. The findings of this study are restricted to the PBMC-EXs derived from SCLC patients, exhibiting a heterogeneous modulation of oncogenic pathways, which reflects the multifaceted interplay between immune and cancer cells. Such interactions are dependent on both the intrinsic factors related to the PBMC-EXs—which may vary between donors—and the diverse phenotypic characteristics of SCLC cells. The findings of this study should be extended in future research, with additional studies incorporating other models and cell types. As an example, we are aware that we hypothesized that the mechanism underlying the communication between exosomes and tumor cells may be their incorporation, as suggested in the literature, and this aspect could be explored in future studies.

Nevertheless, this study provides a proof of concept that paves the way for new research approaches to PBMC-related biomarkers. In light of the findings of our study, we put forward the hypothesis that PBMC-EX screening may be considered for future studies as a biomarker of tumor response and for the evaluation of therapeutic treatments by clinicians. The use of this method as a noninvasive monitoring and prediction tool for patients would be a valuable and promising avenue for further research.

## 5. Conclusions

In conclusion, our study provides evidence that the isolation and analysis of PBMC-EXs from SCLC patients, correlated with tumor stage and treatment, is feasible and may have further development in future studies as potential biomarkers of the response to therapy. We provided proof of concept that PBMC-EXs are able to modulate in vitro tumor features and impair SCLC cell proliferation and are involved in immune response, probably acting as a shuttle of innate immune DNA/RNA sensors, thus opening the way to subsequent studies on innovative therapeutic strategies with clinical implications.

## Figures and Tables

**Figure 1 cancers-16-03151-f001:**
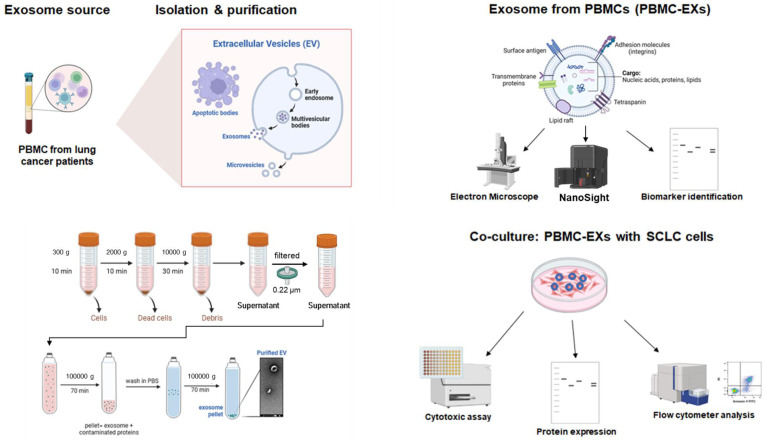
Schematic representation of the exosome isolation protocol, exosome characterization procedure and co-culture with SCLC cell lines. Exosomes were isolated from PBMCs derived from SCLC patients by multiple ultracentrifugation steps. PBMC-EXs were characterized by SEM, nanoparticle tracking analysis was performed using a NanoSight Instrument and exosomal markers were determined by Western blot analysis. The graphical scheme was produced by the authors using the BioRender platform (https://www.biorender.com/) (basic license terms).

**Figure 2 cancers-16-03151-f002:**
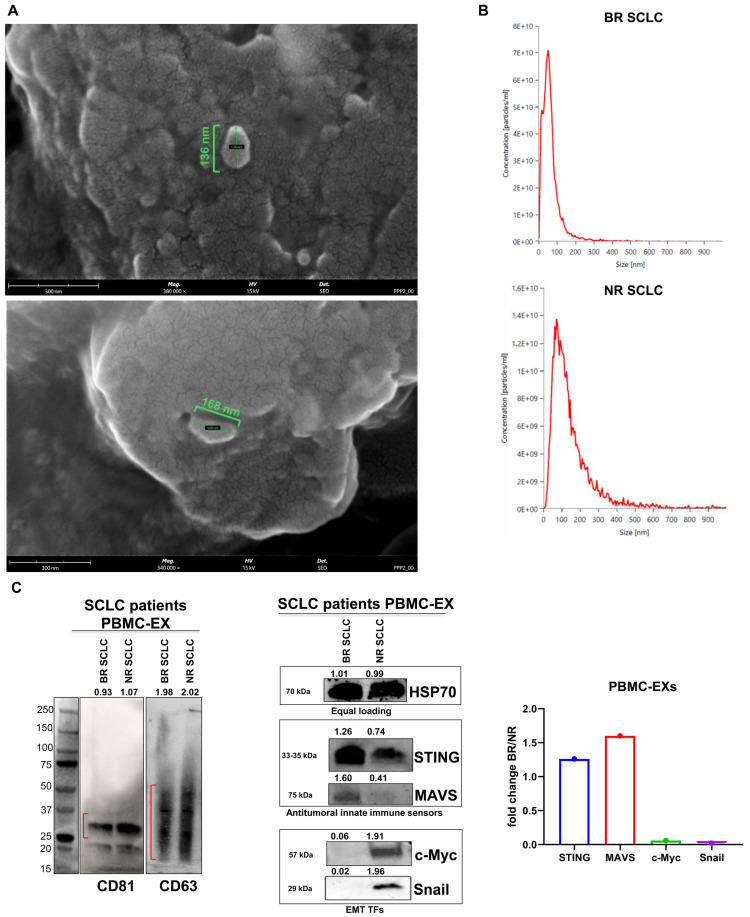
Isolation and characterization of exosomes from SCLC patients PBMC-EXs. (**A**) Representative FEG-SEM images of the isolated exosomes (scale bar = 300 nm). (**B**) Size distribution of the isolated PBMC-EXs. (**C**) Western blot analysis and its quantification of exosomal markers CD81, CD63, HSP70, DNA/RNA sensors of antitumor innate immune response (STING and MAVS) and EMT TFs Snail and c-Myc. The isolated exosomes were successfully isolated from the culture supernatants of PBMCs isolated from BR or NR SCLC patients. The data of PBMC-EX samples were expressed as the ratio of each PBMC-EX BR sample to the corresponding PBMC-EX NR sample to evaluate the relative fold-change induction. Red line indicated the band of each protein on the gel. Original western blots are presented in Appendix A.

**Figure 3 cancers-16-03151-f003:**
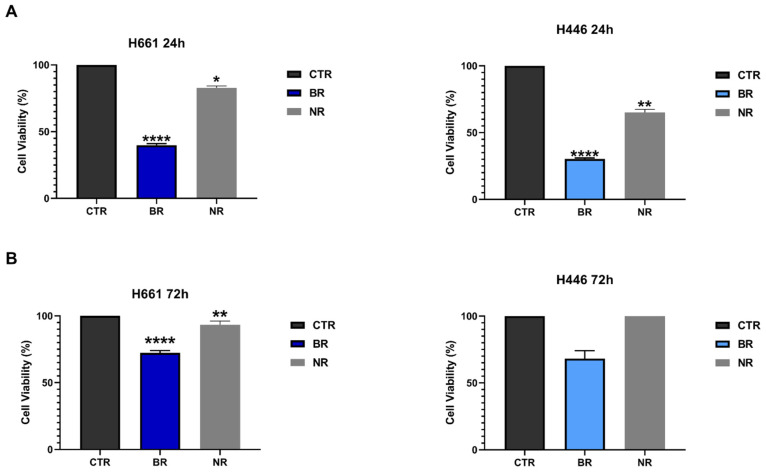
Co-culture effect of PBMC-EXs with SCLC cell lines on cell viability. (**A**) Cell viability of H661 and H446 cells after co-culture for 24 h with PBMC-EXs from BR and NR SCLC patients. (**B**) Cell viability of H661 and H446 cells after co-culture for 72 h with PBMC-EXs from BR and NR SCLC patients. Data are expressed as the mean ± SD. Unpaired Student’s *t*-test with * *p* < 0.05; ** *p* < 0.01; **** *p* < 0.0001.

**Figure 4 cancers-16-03151-f004:**
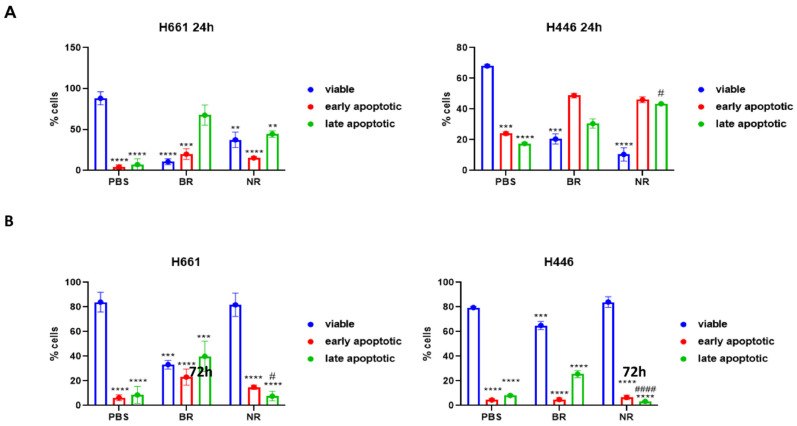
Co-culture effect of PBMC-EXs with SCLC cell lines on cell death. (**A**) Flow cytometry analysis of cell death by Annexin V/PI assay after co-culture for 72 h of PBMC-EXs from BR and NR donors. (**B**) Bar graph showing summary data of % Annexin V/PI positive cells; H661 (upper panel) and H446 (lower panel). Statistical significance: **** *p* < 0.0001, *** *p* < 0.001, ** *p* < 0.01, # = comparison between H661 and H446 apoptosis; #### *p* < 0.0001.

**Figure 5 cancers-16-03151-f005:**
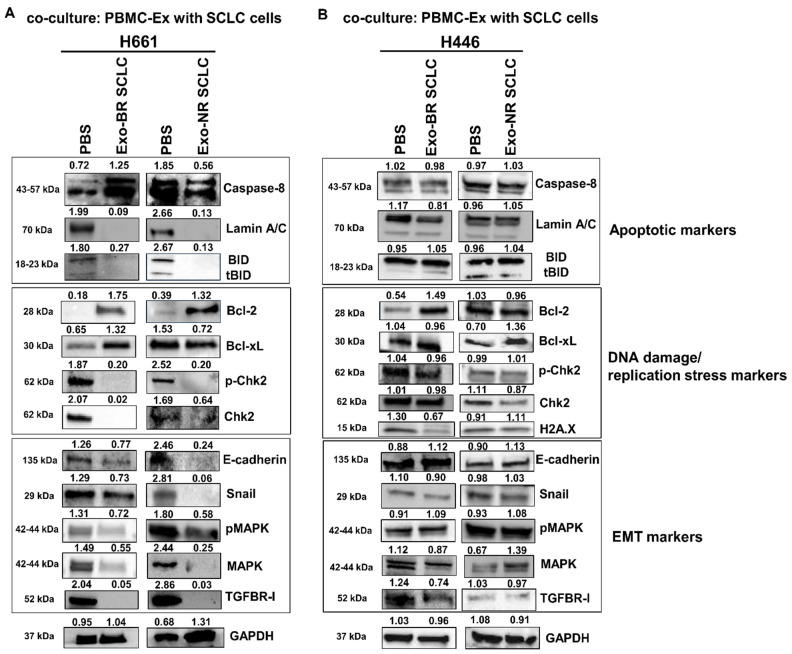
Apoptotic markers, DNA damage/replication stress markers and EMT markers in co–cultures of SCLC cell lines with PBMC–EXs derived from BR and NR SCLC patients. Representative Western blotting of whole cell lysates from (**A**) H661 and (**B**) H446 cell lines showing levels of apoptotic markers (caspase 8, lamin A/C, BID/tBID), DNA damage, replication stress markers (Bcl–2, Bcl–xL, p–Chk2/Chk2, H2A.X) and EMT markers (e–cadherin, Snail, pMAPK/MAPK, TGFBR–I) after co-culture with BR or NR PBMC-EXs. GAPDH was used to ensure equal loading. At least three independent experiments were performed. Original western blots are presented in Appendix A.

**Table 1 cancers-16-03151-t001:** Patient characteristics.

	1L-IO ^1^ (n = 4)	ICT ^2^ (n = 3)	CT (n = 3)
Patients (n)			
BR	3	2	3
NR	1	1	0
Age (mean, range)	63.50 (54–79)	67.00 (54–79)	60.33 (56–64)
Sex (n, %)			
Female	-	1 (33.33%)	1 (33.33%)
Male	4 (100.00%)	2 (66.67%)	2 (66.67%)
Histology (n, %)			
Limited Stage SCLC	-	-	3 (100.00%)
Extensive Stage SCLC	4 (100.00%)	3 (100.00%)	-

^1^ 1L-IO: patients receiving PD-(L)1 inhibitors as a monotherapy in the first line. ^2^ ICT: patients receiving chemotherapy plus PD-(L)1 inhibitors in the second or subsequent lines. 1L: first line; IO: immunotherapy; ICT: chemoimmunotherapy combination; CT: platinum-based chemotherapy; PD-L1-inhibitors: atezolizumab, durvalumab. SCLC: small cell lung cancer.

**Table 2 cancers-16-03151-t002:** Quantification of PBMC-EX protein content (µg per 12 mL of blood) according to clinical features (tumor stage and response to chemoimmunotherapy).

Patients (n = 6)	PBMC-EX Yield *^#^
(LS)-SCLC	<12 μg
(ES)-SCLC BR	1520.2 μg ± 232
(ES)-SCLC NR	4392.2 μg ± 1128

* Blood amount = 12 mL ± 0.5, ^#^ Number of PBMCs = 20 × 10^6^ ± 10.

## Data Availability

The majority of data related to the presented results are included in the Materials and Methods section of this paper.

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
