# Peer review of "Immune-Cell-Derived Exosomes as a Potential Novel Tool to Investigate Immune Responsiveness in SCLC Patients: A Proof-of-Concept Study"

_cancers, 2024, doi:10.3390/cancers16183151_

Round 1

Reviewer 1 Report

Comments and Suggestions for Authors

Keywords should appear in alphabetical order. The introduction is very short and should be more extensive explaining a little more background. Figures 2 and 4 have graphs, and these should appear separately and be called graphs and not figures.

Comments on the Quality of English Language

The article should be publish after minor revision

Reviewer 2 Report

Comments and Suggestions for Authors

The manuscript Amato et al. reported a proof-of-study on the isolation and characterization of EVs from small cell lung cancer patients’ immune cells. During the study, the authors compared the cases for patients with best response and non-response to chemo-immunotherapy. Majority of effects were on the comparison of cellular response of isolated EVs. However, in the reviewer’s option, it was more important to better characterize the EVs from both cases in terms of size, size distribution, and biomarkers.

1.       The author presented SEM images of the EVs, which did not provide much information on the size, size distribution and morphology of the EVs. Instead, TEM or cryo-TEM images need to be provided to illustrate the isolated EVS.

2.       Mean sized from NTA were provided in a table. Size plots are needed to show the size and size distribution.  The size and size distribution based on NTA analysis are highly dependent on the purity of the isolation, because the light scattering mode of NTA cannot differentiate EVs from other nanosized objects, such as protein aggregate, salt crystals, etc. Fluorescent mode NTA is strongly suggested to demonstrate the presence of key protein makers and show the biomarker differences in the two cases.

3.       Characterization of typical EV markers are needed, such as CD63, CD81, CD9, ALIX, along with the specific marker associated with small cell lung cancer.

Reviewer 3 Report

Comments and Suggestions for Authors

First, in my opinion, the title of the manuscript does not reflect its content. The study described in the manuscript was not devoted to the simple isolation and characterization of exosomes derived from immune cells of patients with small cell lung cancer, but concerned the comparison of exosomes obtained from SCLC patients with different responses to chemo-immunotherapy (BR and NR groups). It is also good practice to annotate the results obtained in the title.

The purpose of the study is presented very enigmatically and somewhat confusingly. Contrary to the statement made in sentence lines 81-83, the research methodology presented in the manuscript does not, in my opinion, in any way address the possibility of using EVs for diagnostic or therapeutic purposes. The second sentence actually describes the objectives pursued more, but it would be worthwhile to make it even more specific. No potential biomarkers have been identified, and the analysis of the expression of selected proteins performed is semi-quantitative. The results obtained require verification by mass spectrometry.

Contrary to the title, section 2.1 does not contain any information about the patients included in these studies. All that is known is that an unspecified number of patients were divided into two groups BR and NR. It is not known how many patients in the BR group were exclusively treated by chemotherapy (cisplatin), how many by an anti-PD-L1 antibody (atezolizumab, durvalumab), and how many had these two treatments.

Analyzing the isolation protocol (section 2.4), I had serious doubts that exosomes were actually isolated. The two initial centrifugations used (2000 x g, 10 min and 10000 x g, 30 min) only allow the removal of cells, cellular debris and apoptotic bodies. However, microvesicles remain in the sample, as centrifugation at 16000-18000 x g is needed to pelleted them. Thus, another centrifugation at 100000x g yields a pellet that is a mixture of exosomes and microvesicles. This is confirmed by the results obtained from NTA (Fig. 2B), where the EV mean is 135± 66 nm for PBMC-EX BR and 200±31 nm for PBMC-EX NR. Exosomes are assigned a much smaller diameter. For documentation of the results from NTA, it would still be useful to have a histogram to see the actual diameter distribution of the isolated EV sample.

As for the semi-quantitative analysis of protein expression in H661 cand H446 ells (Section 2.7) after their incubation with EV samples, it would still be useful to show that indeed EVs are incorporated by these cells.

I have not yet encountered such an approach to analyze the amount of protein in EVs (section 3.2). Analysis of their proteome focuses on qualitative and quantitative parameters.

Fig. 5: no semi-quantitative analysis, analogous to Fig. 2C. It is difficult to judge from the image of representative blots alone whether the description of the results is correct.

The statement that '(we) analyzed their protein cargo- line 392' is over the top. No qualitative or quantitative analysis of the EV samples' proteome was conducted.

Minor comments:

line 133: should be: Bradford assay

line 167: instead of rpm please give rcm

line 168: What exact amounts of proteins were applied to the wells?

Linne 225: Immunodetection of only one of the exosome markers, namely Hsp70, was performed.

Reviewer 4 Report

Comments and Suggestions for Authors

This is an interesting study about the isolation and characterization of exosomes from PBMC of patients resposive to therapy or non-responsive, however, there are lot of open questions not addressed in this paper with the discussion being the weakest part.

1.       Introduction: The authors cite the paper Small Cell Lung Carcinoma: Current Diagnosis, Biomarkers, and Treatment Options with Future Perspectives (by Krpina et al) stating „For decades, platinum-based chemotherapy regimens have constituted the single option of therapy for  patients with SCLC, with a five-year overall survival (OS) rate of only 10%“, but in the cited paper Krpina and other authors state: „While SCLC exhibits initial responsiveness to chemotherapy and radiotherapy, treatment resistance often emerges, leading to a five-year overall survival rate of only 10%.“

Is it just chemotherapy or also radiotherapy?

2.       Line 95 I am not sure what „as the best response“ means here

3.       Line 117 what is the producer for exosome depleted FBS ?

4.       Figure 1. This figure needs to be explained in figure legend. This is only the title.

5.       2.6. Co-colture protocol-> co-culture protocol (there is more of this mistake throughout the text and figures

6.       Line 201 Manufacturer of flow cytometer?

7.       Figure 2A-> the designation of the size on SEM images must be larger. Now you cannot see it.

8.       Please add original figures from NanoSight instrument to Figure 2B. Only the table is not a result.

9.       On western blots the size of every protein in kDa should be added.

10.   Figure 2C. The expression of HSP70 is not a good marker for exosomes, as it can also be expressed in normal cells, especially tumor cells and in conditions of stress. There are better markers for exosomes, such as Tetraspanins (e.g. CD63, CD81, CD9). The authors should also present a control protein that is only present in normal cells, but absent from exosomes (such as calnexin). HSP70 is and OK loading control. Please complete the figure as suggested.

11.   Figure 2C. Fold change PBMC-EX BR/NR-> I am not sure that this is the best way to show the results, because this way it seems like a difference in the expression between BR and NR in Snail/Myc is minimal when in fact it is huge as in BR it is not expressed and in NR it is highly expressed.

12.   Line 305 flow cytometry staining-> the staining is performed by annexin/PI, and by flow cytometry method the cells are analyzed.

13.   Lines 312-313-> „In addition, it was observed that H661 cells exhibited greater sensitivity to apoptosis induced by PBMC-EXs, than H446 cells.“ In order to claim this, the authors should provide the statistics comparing apoptotic H661 cells compared to apoptotic H446 cells.

14.   Lines 331-332 After 72 h of incubation, protein extraction and western blot analysis were performed. -> Why did the authors incubate the cells for so long to analyze proteins, when usually 24-48 hours is enough?

15.   I have a lot of questions about Figure 5 A and H661 cell line which had a significant amount of cells in apoptosis after the treatment with BR exosomes:

·         Since Bid is a pro-apoptotic protein, why is it reduced in H6611 exo-BR (and also NR) treated cells?

·         Why are Bcl2 and Bcl-xl increased in H661 Exo-BR treated cells (as they are anti-apoptotic proteins)?

·         Why are p-MAPK and MAPK reduced in H661 non-responders?

The authors do not comment these results, but merely say „Whereas, H661 cells, did not show significant changes in these markers (Figure 5A), thus suggesting intrinsic cell phenotype may affect these responses.“, which is not really correct. The authors need to address these results in the cell line which actually showed apoptotic cells.

I would rewrite this part of the results and focus more on the significant results from the H661 cell line in which significant apoptosis was induced following the treatment with BR exosomes.

16.   Figure 5- again co-colture

17.   Line 401 induce cell toxicity is not a proper expression in this context

18.   The discussion is too short and it does not address important questions of this paper nor does it state any limitations of the study. For example, what exactly can be found in exosomes to induce cell death (apoptosis)? Or what is the cause for different protein expression in these exosomes? One of them is also the role of these different exosomes in immune system response which has not been addressed at all in the discussion. Also, are the authors first to use these exosomes for diagnostic (biomarkers) and/or  for therapeutic reasons, otherwise why would they call it proof of concept? But they did not mention this anywhere? The authors need to rewrite the discussion with emphasis on the important open questions.

Comments on the Quality of English Language

No big issues detected.

Round 2

Reviewer 3 Report

Comments and Suggestions for Authors

Comments on authors' responses

After carefully reviewing the authors' responses to my comments and the reviewed version of the manuscript, my comments are as follows:

 1. I don't think the new title of the manuscript reflects well the results described in it. What kind of molecular changes do the authors have in mind? There was no analysis of the EVs cargo composition. It seems to me that the issue of the study was more about the possibility of predicting immune responsiveness of SCLC patients, which is what the authors write about in the abstract (lines 39-42). If there are any specific proteins involved, it should be clearly specified.

 2. The research objective formulated in the manuscript (lines 114-119) is incorrect. No analyses of the EVs proteome composition were conducted. This is surprising, because in response to one of my comments, this objective was properly defined. Quote: We specified that this paper is a proof-of-concept study to use exosomes, from immune cells of SCLC patients under treatment, as a tool to detect tumor resistance or sensitivity to chemo-immunotherapy and to explore their role as cytotoxic agents.

 3. The first version of the manuscript did not mention that between centrifugation at 10,000 x g and 100,000 x g there was an additional step, that is, filtration of the supernatant using 0.22 μm filter. If this was indeed during the procedure for isolating the EVs, I consider it a very significant omission. This applies to both section 2.3 and 2.4.

Besides, using a 0.22 μm filter does not guarantee that all EVs larger than 200 nm will be removed. And this can actually be seen in Figure 2. As for distinguishing between exosomes and microparticles, the size ranges (30-100 nm and 100-1000 nm) are highly arbitrary - exosomes and micropartices probably overlap in size, especially around 100±50 nm. Since in the isolated sample of EVs most of the objects are less than 200 nm in diameter, it would be better to use terms such as "small EVs (<200 nm) and "large EVs" (>200 nm). The two different EV-related terminologies (exosomes/microvesicles vs. small/large EVs) are consistent with the latest "Minimal information for studies of extracellular vesicles (MISEV2023)" guidelines. For this reason, I recommends changing the term “exosomes’ to ‘small EVs” in the manuscript. In most cases, exosomes are considered to be up to 150 nm in diameter, unless the cellular origin (endosomal vs. outer cell membrane) can be clearly demonstrated for an isolated sample.

 4. Showing that indeed EVs are incorporated by H661 cand H446 cells on the basis of protein level is not possible at all. For that you need confocal microscopy. I agree that such images cannot be taken within a week, which does not change the fact that it should be done earlier.

 5. I remain of my opinion regarding an approach to analyze the amount of protein in EVs (section 3.2). Examples of two articles representing this approach are too few.

 6. Semi-quantitative analysis of the results shown in Figure 5 is possible without mass spectrometry. This should be done analogously as for the results shown in Fig. 2C. Especially since Section 2.10 states that densitometric analysis of the blots was performed.

New comments regarding the revised manuscript

1. Simple abstract by definition should be a simplified version of Abstract, which is not the case here. Also, a correction is needed: (1) according to the results described in the manuscript, the characterization of the PBMC-EXs protein content was not performed, (2) what are the MYC markers (line 36), other than ETM markers. They are not listed in the Abstract., (3) a higher level of apoptosis (line 38) line or late apoptosis(line 53).

 2. The study included 10 patients, but the number of subjects in the BR and NR groups was not even similar (8 vs. 2 subjects).

 3. In Figure 1 Isolation & filtration – lack on the diagram filtration step by 0.22 μm filter.

 4. In Figure 1 Exosome from PBMCs – lack of  pictogram showing Nanoparticle Tracking Analysis

 5. In Figure 1 Coculture…: instead of Cytotoxic assay should be MTT assay, instead of Protein expression should be Western blot, instead of Flow cytometry analysis should be Apoptotic assay.

 6. Figure 1 legend: Legend to be clarified. (1) What exactly does co-culture system mean? Other parameters were evaluated after running co-culture. (2) What kind of "PBMC-EX specificity" were the authors thinking of? I think they were referring to identification of EX markers by WB. (3) It seems to me that the use of graphics prepared in BioRender platforms for publication requires a license.

 7. Figure 1A: Font of numbers regarding scale bar and diameter: too small, poorly legible. Needs to be enlarged.

 8. Line 175: Nanosight analysis or Nanoparticle Tracking Analysis?

 9. Line 268: performed by Nanoparticle Tracking Analysis rather than performed with a NanoSight instrument.

 10. Figure 2 legend: (1) What is actually the scale bar? Only 300 nm appears on panel A. (2) Contrary to the statement Data of 310 PBMC-EX samples were expressed as the ratio of each PBMC-EX BR sample on the corresponding 311 PBMC-EX NR sample to evaluate the relative fold change induction, no (semi-)quantitative analysis is shown in Figure 2. In the previous version of the manuscript, such a semi-quantitative analysis was shown.

 11. The information presented in Table 2 should be included in Figure 2B.

 12. Figure 3: Too many of these panels, the chart is not very clear. Since, anyway, for each EX (derived from BR and NR PBMC) and each time (24 h and 72 h) each time the results were scaled against the control (taken as 100%), all the results for the mentioned conditions can for further cell line be put on one chart.

 13. Figure 4D: no information on the panel that the results are for 72 h. Analogous to what is shown in panel B.

 14. Line 447: not protein content but protein expression.

 15. Figure 5: lack of results showing the semi-quantitative analysis of Western blots.

Reviewer 4 Report

Comments and Suggestions for Authors

I did not find the results for the calnexin.even if negative, it should be presented.

Comments on the Quality of English Language

English is fine.

Round 3

Reviewer 3 Report

Comments and Suggestions for Authors

The authors responded to all my comments mentioned in my review and brought to the manuscript the necessary changes that would definitely enhance its quality.